# Solvent Moisture-Controlled Self-Assembly of Fused Benzoimidazopyrrolopyrazines with Different Ring’s Interposition

**DOI:** 10.3390/molecules27082460

**Published:** 2022-04-11

**Authors:** Svetlana V. Martynovskaya, Arsalan B. Budaev, Igor A. Ushakov, Tatyana N. Borodina, Andrey V. Ivanov

**Affiliations:** A. E. Favorsky Irkutsk Institute of Chemistry, Siberian Branch of the Russian Academy of Sciences, 1 Favorsky St., 664033 Irkutsk, Russia; svetakuz@irioch.irk.ru (S.V.M.); arsalan@irioch.irk.ru (A.B.B.); igor-papa71@mail.ru (I.A.U.); borodina@irioch.irk.ru (T.N.B.)

**Keywords:** aldehydes, allenes, cyclization, heterocycles

## Abstract

This article shows that two extremely important families of fused heterocyclic assemblies, namely 6-methylbenzo[4,5]imidazo[1,2-*a*]pyrrolo[2,1-*c*]pyrazine and 5a-methyl-5a,6-dihydro-5*H*,12*H*-benzo[4,5]imidazo[1,2-*a*]pyrrolo[1,2-*d*]pyrazine, can be synthesized from only two available building blocks (*N*-allenylpyrrole-2-carbaldehyde and *o*-phenylenediamine) by controlling only one reaction parameter (water content of the medium). It should be emphasized that the latter class of compounds (with an *a/d* arrangement) is previously unknown. If the allene group is introduced not into the starting compound, but during the reaction (in superbase media), a heterocyclic ensemble, 5-methylbenzo[4,5]imidazo[1,2-*a*]pyrrolo[2,1-*c*]pyrazines, with a different position of the methyl group is formed.

## 1. Introduction

Fused heterocyclic compounds, namely, derivatives of imidazole, benzimidazole, and pyrazine, exhibit diverse biological activities. Imidazolopyrazines and imidazolopiperazines belong to the class of therapeutically efficient fused heterocycles. Imidazolopyrazine derivatives exert anti-inflammatory and antiviral actions, as well as inhibition MAPK-activated PK5 [1]. It was reported that imidazolopyrazines act as an effective CXCR3 antagonist (regulation of leukocyte transport) [2], and as a potent IGF-1R inhibitor [3]. The literature search reveals that the imidazolopiperazine motif is responsible for antimalarial action of known drugs [4], which can be enhanced via the introduction the pyrrole moiety in their structure.

Despite a wide applicability of such compounds, facile and straightforward methods for the synthesis of these biologically active scaffolds still remain limited. Currently, only a few examples of assembling such systems are documented in the literature. They can be divided into two groups: intramolecular and intermolecular reactions (Figure 1). Polyheterocyclic systems are obtained via intramolecular cyclization in the presence of palladium [5] or copper [6,7] salts as catalysts at high temperatures. Intermolecular cyclizations are also promoted by catalysts: transition metals [8], salts [9] or acids [10]. All the above reactions require expensive toxic catalysts, long reaction times or high temperatures, which are disadvantages from a synthetic point of view.

## 2. Results

We have developed a strategy for the synthesis of benzimidazopyrrolopyrazines through the sequential addition of *o*-phenylenediamine (*o*-PDA) **2** to *N*-allenylpyrrole-2-carbaldehydes **1a**–**j** under mild conditions. Our investigation commenced with the reaction of 5-(2-phenyl)-*N*-allenylpyrrole-2-carbaldehyde **1b** and *o*-PDA **2**, which proceeded under conditions developed by us earlier for *N*-vinylpyrrole-2-carbaldehydes [11] in ethyl alcohol at room temperature for 16 h (Figure 2).

In contrast to the work with N-propargylindolecarbaldehyde [8], the allene fragment does not require activation with heavy metal salts (CuI), which opens up more possible directions for the reaction.

Experiments have shown that the reaction leads to a mixture of two products: the anticipated benzimidazopyrrolopyrazine **3b** and the unexpected dihydrobenzimidazopyrrolopyrazine **4b**. The process is influenced by the air humidity and hence by the presence of some amount of water in the reaction mixture. Thus, at a higher content of water in the air (wet season), the ratio of products **3b** and **4b** is 1:1 (NMR), while in a drier climate in the presence of commercial ethanol, this ratio is 64%:16% (19% of unreacted compound **1b**). Consequently, we have optimized this reaction by the addition of water and varying the solvents in order to direct it selectively to one of the two products (Table 1).

Optimization of the reaction conditions has revealed that in commercial DMSO, which usually contains some water, the process occurs with similar efficiency to that observed in aqueous ethanol (Entry 2). When the reaction was carried out in DMSO at a higher temperature (65 °C) for a shorter time (1 h), the products were obtained in the same ratio (Entry 3). However, in this case, the reaction was accompanied by the formation of hardly identifiable side products and afforded the target products in lower yield. Therefore, we further used the initial temperature conditions. Subsequently, we carried out experiments with dried ethanol in an inert atmosphere to avoid the effect of air humidity. In dry ethanol, a mixture of two products **3b** and **4b** was formed in a 9:1 ratio. Evidently, the dried ethanol contained enough water to trigger a side reaction (Entry 5). In the presence of ethanol and 10% of water, a mixture of two compounds **3b** and **4b** was obtained in a 6:4 ratio (Entry 6). At a higher content of water (20%), the product ratio was again 1:1 (Entry 7), which evidenced the clear dependence of this ratio on the amount of water. However, when the process was implemented in a 1:1 mixture of ethanol-water (Entry 8), the starting compound **1b** was completely recovered due to poor solubility of such a mixture. Replacement of ethanol with methanol, which usually contains much less water, led to almost selective formation of product **3b** (Entry 9). The reaction in commercial (Entry 10) and dry benzene (Entry 11) also resulted in a higher selectivity with respect to compound **3b**, the conversion in the latter being decreased.

Thus, to selectively synthesize compounds **3**, it is most expedient to employ the reaction conditions shown in Entry 9. Under these conditions, a wide series of N-allenylpyrrole-2-carbaldehydes was involved in the process to selectively afford benzimidazopyrrolopyrazines **3a**–**j** (Figure 3).

We failed to selectively direct the reaction toward the formation of products **4**. Therefore, aqueous ethanol (Figure 2) was used, followed by the isolation of individual compounds by column chromatography. The structures and yields of the products obtained by both methods are shown in Table 2.

As can be seen from Table 2, aromatic and heteroaromatic substituents in the α-position of the pyrrole ring have the same effect on the yield of the reaction products. This is also the case for donating or accepting substituents in the para position of the phenyl substituent. A slight decrease in yields for bulky compounds **4h** and **4i** is due to the steric hindrance. The decreased yield and incomplete conversion in the case of starting pyrrole **1a** bearing donor alkyl substituents is expected for nucleophilic addition at the carbonyl group, since the donating substituents compensate the positive charge on the carbonyl carbon.

The structure of benzoimidazopyrrolopyrazines **3** was unambiguously proven by X-ray diffraction analysis using compound **3e** as an example (Figure 1).

The structure of **4** was established by NMR spectroscopy (^1^H, ^13^C, including 2D correlations, Figure 2a,b.

The formation of products **3** can be tentatively rationalized as follows [12,13]. The reaction starts with the addition of *o*-PDA to *N*-allenylpyrrole-2-carbaldehyde to furnish Schiff base **A**. The latter further undergoes intramolecular cyclization to give the benzimidazole skeleton **B** (*5-exo-trig*). This is a well-known reaction, in which ambient oxygen can act as an oxidizing agent for the intermediate imidazoline **B**. The *NH*-function of the benzimidazole attacks the central carbon atom (*sp*) of the allene moiety that leads to a *6-exo-dig* cyclization to finally deliver benzimidazopyrrolopyrazines **3** (Figure 4).

It is more difficult to explain the formation of product **4**. We have proposed two possible reaction pathways.

Possible pathway 1: The formation of by-product **4** likely also commences with the addition of *o*-PDA to *N*-allenylpyrrole-2-carbaldehyde to produce the Schiff base **A**. Afterward, the second amino group is not added to the C=N bond (as in Figure 4), but at the central allenic carbon to form a nine-membered ring **D** (Baldwin’s rule [14,15,16]). Product **4** contains two more hydrogen atoms than compound **A**, i.e., some kind of reduction occurs. Apparently, the intermediate imidazoline **B** generated during the formation of compound **3** with the hydrogen transfer through the medium can act as a reducing agent. In the reduced intermediate **E**, the double bond is activated by the acid of the system to intramolecularly close the ring and to afford product **4** (Figure 5).

As seen from the above mechanistic schemes, selectivity of the reaction depends on the behavior of the amino group in intermediate **A**, which attacks either the C=N or C=C bond. A possible reason for the effect of water on the reaction direction is assumed to be the presence of hydrogen bonding between water molecules and the amino group, which complicates the attack on the more sterically hindered C=N bond thus allowing an alternative addition at the C=C bond. This scheme is supported by the fact that the best ratio of products of **3:4** was 1:1 (Table 1, Entry 7); compound **4** cannot be formed without a reducing agent (intermediate **B**) delivering product **3**. Besides, a further increase in the water content to direct the reaction selectively to product **4** had no success due to the reduction of solubility of the starting compound in the water–alcohol mixture (Table 1, Entry 8).

Possible pathway 2: It can also be assumed that the reaction is triggered by disproportionation of the starting pyrrolecarbaldehyde **1** (the Cannizzaro type reaction [17]) to furnish carboxylic acid **F** and alcohol **G**, which further interact with *o*-PDA. In the case of carboxylic acid, the process involves the formation of product **3** similar to the reaction with aldehyde (such reactions are well known [18,19,20]). In alcohol **G**, the substitution at the hydroxyl group occurs with participation the *o*-PDA amino group to deliver the intermediate **H**, followed by intramolecular addition of the second amino group at the allene producing a 9-membered ring **I**. The latter can undergo intramolecular cyclization due to the activation of the remaining double bond by the acid of the system to afford compound **4** (Figure 6).

We indirectly proved such direction of the process by involving the separately obtained alcohol in the reaction. Among the reaction products, an intermediate compound was identified and characterized by NMR spectroscopy (see Appendix A). The obvious disadvantage of such an explanation of the possible mechanism is that upon disproportionation, the ratio of products should be close to 1:1, but in our case, compound **3** most often predominates. This can be rationalized by the fact that the starting pyrrolecarbaldehyde **1** is not only disproportionated, but also directly reacts with *o*-PDA (Figure 2), thereby increasing the content of product **3** in the mixture. Probably, water here directly affects the direction of Cannizzaro reaction and somehow promotes the hydrogen transfer from the oxidizing site to the reducing one thus enhancing the contribution of Cannizzaro-type reaction to the overall process. This also explains the impossibility to shift the ratio of products **3** and **4** by more than 1:1. Similar to the case of pathway 1, a further increase in the concentration of water decreases solubility of the reactants and worsens the outcome of the reaction.

Non-conjugated dihydrobenzimidazopyrrolopyrazines **4** represent a rather rare class of compounds. To the best of our knowledge, chemical databases contain no such structures. We managed to find a structure with a similar 5/6/5 arrangement at the *a*/*d* position, containing only one nitrogen atom and synthesized by the Pauson–Khand reaction [21,22,23], all the obtained compounds being considered as promising alkaloids. Meanwhile, these compounds are expected to exhibit valuable biological properties and can be widely employed as synthons in organic synthesis.

Often, the relative position of substituents in a molecular backbone and the presence of free positions suitable for further functionalization are essential for biological applications. In all disclosed protocols for the assembly of benzimidazopyrrolopyrazines **3**, the spacer between the pyrrole/indole nitrogen and the imidazole ring nitrogen either does not bear a methyl substituent at all, or contains a methyl moiety in the *α*-position relative to the imidazole ring. Synthesis of benzimidazopyrrolopyrazines with the methyl group located closer to the pyrrole/indole fragment will expand the synthetic potential, which possibly complements the existing backbones to afford new biological targets.

To address this challenge, we have employed the following synthetic approach based on our previous research (Figure 7). At the first stage, a series of *NH*-pyrrolylbenzimidazoles **5** were prepared via the published procedure [11]. Further, the obtained compounds were involved in the reaction with propargyl chloride under the conditions reported earlier [24]. The superbase in this reaction has an advantage over the common bases: it catalyzes not only substitution of the *NH*-function with propargyl chloride, but also the complete (quantitative) acetylene-allene isomerization to furnish allene, which is almost always selectively attacked at the *sp* atom. Since in compound **5**, the *NH*-function of imidazole (pKa = 14.4) is more mobile than that of pyrrole (pKa = 23.0), substitution occurs on the former. Product **I** undergoes isomerization into intermediate **J**, the *sp* atom of which is subjected to an intramolecular attack by the pyrrole *NH*-moiety and *6-exo-dig* cyclization to deliver a new series of benzimidazopyrrolopyrazines **6**, isomeric to compounds **3**, but with a different disposition of the methyl group. We checked the generality of this strategy on a series of substrates, and the product yields are shown in Table 3.

We checked the generality of this strategy on a series of substrates, and the product yields are shown in Table 3.

The reaction was found to be of general character. It was shown that aromatic and heteroaromatic substituents did not affect yields of the products (**6a**–**i**, **k**–**l**). If the *p*-position of the phenyl ring contained donating (**6d**, 90%) or accepting substituents (**6g**, 96%), the yields were approximately the same. In the case of a donating substituent in the pyrrole ring (**6l**), the yield decreased to moderate.

The structure of the obtained compounds was unambiguously proven by X-ray single-crystal analysis using **6b** as an example (Figure 3). For all the processes described above (Figure 2 and Figure 3; Table 3), at each stage of several possible (allowed) directions, according to Baldwin’s rule, only one is always realized, i.e., all reactions are regeoselective.

## 3. Materials and Methods

### 3.1. General Information

All reagents were purchased from a commercial supplier, such as Sigma-Aldrich Co. LLC (St. Louis, MO, USA). *N*-Allenylpyrrole-2-carbaldehydes were obtained according to the procedure [25]. DMSO was used with a water content of 0.1–0.3%, ethyl alcohol with a water content of 5%. Methyl alcohol used 99.5%. Ethanol and benzene were dried according to the literature data [26]. Propargyl chloride was purified by simple distillation (58 °C). NMR spectra were recorded from solutions in CDCl_3_ and DMSO-*d_6_* on Bruker DPX-400 and AV-400 spectrometers (400.1 MHz for ^1^H and 100.6 MHz for ^13^C). Chemical shifts (δ) are quoted in parts per million (ppm). The residual solvent peak, δ_H_ = 7.27 and δ_C_ = 77.16 for CDCl_3_, δ_H_ = 2.50 and δ_C_ = 39.52 for DMSO-*d_6_*, was used as a reference. Coupling constants (*J*) are reported in Hertz (Hz). The multiplicity abbreviations used are: s, singlet; d, doublet; t, triplet; q, quartet; m, multiplet; br, broad signal.

### 3.2. General Procedures

General procedure for the synthesis products **3** and **4**. A mixture of the *N*-allenylpyrrole-2-carbaldehyde 1 (0.0011 mol), *o*-PDA 2 (0.131 g, 0.00121 mol), EtOH (2.2 mL), H_2_O (0.44 mL) and CF_3_COOH (1%) was stirred at r.t. for 16 h. Solvent was evaporated. The resulting products were separated on a column with Al_2_O_3_ using flash chromatography (eluent hexane then hexan/diethyl ether 1:1).

General procedure for the synthesis products **3**. A mixture of the *N*-allenylpyrrole-2-carbaldehyde 1 (0.0011 mol), *o*-PDA 2 (0.131 g, 0.00121 mol), MeOH (dry, 2.2 mL) and CF_3_COOH (1%) was stirred at r.t. for 16 h in an inert atmosphere. The resulting precipitate was collected.

General procedure for synthesis of **5**. Compounds **5** were prepared according to procedure [11]. Compounds **5a**, **5c**–**i**, **5l** were synthesized for the first time. Product yields depend on the complexity of the purification of the raw product. A mixture of *NH*-pyrrole-2-carbaldehyde (0.002 mol), *o*-PDA 2 (0.216 g, 0.002 mol) and CF_3_COOH (1% with respect to the combined mass of both reagents) in DMSO (2 mL) was stirred at 70–80 °C for 1 h with continuous air bubbling. The reaction mixture was diluted with aq 1% NaHCO_3_ soln (8 mL), extracted with Et_2_O (5 × 5 mL) and the extracts were dried (K_2_CO_3_). The solvent was evaporated, and the crude product was passed through a neutral alumina column (eluent hexane, then hexane/diethyl ether (1:1)) to give **5**.

General procedure for synthesis of **6**. A mixture of **5** (0.0011 mol), KOH pellets (0.286 g, 0.0044 mol) and DMSO (water content < 0.2%) (2.2 mL) was stirred at r.t. for 45 min. Subsequently, freshly distilled propargyl chloride (0.164 g, 0.0022 mol) was added over 10 min, while keeping the internal temperature between 28 and 30 °C (exothermic reaction). A further amount of KOH pellets (0.858 g, 0.0132 mol) was added, while heating between 35 and 40 °C for 20 min, and then the reaction mixture was poured into H_2_O under efficient stirring. The formed precipitate was filtered off.

### 3.3. Characterization Data of Products **3**,**4**,**5** and **6**

3-Butyl-6-methyl-2-propylbenzo[4,5]imidazo[1,2-*a*]pyrrolo[2,1-*c*]pyrazine (**3a**). White powder (0.078 g, 30% yield; 0.147 g, 42% yield). ^1^H NMR (400 MHz, CDCl_3_) δ 7.86–7.83 (m, 2H, Ph), 7.36 (t, *J* = 7.6 Hz, 1H, Ph), 7.21 (t, *J* = 7.7 Hz, 1H, Ph), 7.13 (s, 1H, CH), 6.92 (s, 1H, pyrrole), 2.83 (s, 3H, CH_3_), 2.79 (t, *J* = 7.6 Hz, 2H, CH_2_), 2.57 (t, *J* = 7.5 Hz, 2H, CH_2_), 1.71–1.66 (m, 2H, CH_2_), 1.61–1.56 (m, 2H, CH_2_), 1.44–1.39 (m, 2H, CH_2_), 1.01–0.95 (m, 6H, CH_3_). ^13^C NMR (100 MHz, CDCl_3_): δ 144.8, 143.5, 131.2, 126.9, 125.6, 123.6, 121.3, 119.7, 119.3, 118.7, 112.5, 107.1, 105.6, 31.3, 28.3, 24.1, 23.8, 22.7, 18.1, 14.1, 14.0. Elemental analysis calcd (%) for C_21_H_25_N_3_: C 78.96, H 7.89, N 13.15; found: C 79.12; H 7.93; N 13.28.

6-Methyl-3-phenylbenzo[4,5]imidazo[1,2-*a*]pyrrolo[2,1-*c*]pyrazine (**3b**). White powder (0.111 g, 34% yield; 0.284 g, 87% yield). ^1^H NMR (400 MHz, CDCl_3_) δ 7.87 (d, *J* = 7.8 Hz, 1H, Ph), 7.82 (d, *J* = 8.3 Hz, 1H, Ph), 7.54–7.47 (m, 4H, Ph), 7.42–7.38 (m, 1H, Ph), 7.37–7.35 (m, 1H, Ph), 7.33 (d, *J* = 4.0 Hz, 1H, pyrrole), 7.24 (s, 1H, CH), 7.22–7.18 (m, 1H, Ph), 6.72 (d, *J* = 4.0 Hz, 1H, pyrrole), 2.73 (s, 3H, CH_3_). ^13^C NMR (100 MHz, CDCl_3_): δ 144.8, 143.5, 131.5, 131.2, 130.8, 129.1 (2C), 128.7 (2C), 128.1, 123.9, 121.9, 121.4, 120.9, 119.6, 112.9, 112.6, 107.5, 106.3, 18.0. Elemental analysis calcd (%) for C_20_H_15_N_3_: C 80.78, H 5.08, N 14.13; found: C 80.81, H 5.12, N 14.17.

6-Methyl-3-(p-tolyl)benzo[4,5]imidazo[1,2-*a*]pyrrolo[2,1-*c*]pyrazine (**3c**). White powder (0.109 g, 32% yield; 0.29 g, 85% yield). ^1^H NMR (400 MHz, CDCl_3_) δ 7.89 (d, *J* = 8.1 Hz, 1H, Ph), 7.86 (d, *J* = 8.4 Hz, 1H, Ph), 7.46 (d, *J* = 8.0 Hz, 2H, Ph), 7.40–7.38 (m, 1H, Ph), 7.36–7.35 (m, 1H, Ph), 7.34–7.31 (m, 2H, Ph, pyrrole), 7.26–7.22 (m, 2H, Ph, CH), 6.71 (d, *J* = 3.9 Hz, 1H, pyrrole), 2.77 (s, 3H, CH_3_), 2.44 (s, 3H, CH_3_). ^13^C NMR (100 MHz, CDCl_3_): δ 144.7, 143.5, 138.1, 131.2, 130.9, 129.8 (2C), 128.6 (2C), 128.5, 123.8, 121.8, 121.1, 120.7, 119.6, 112.6, 112.6, 107.6, 106.3, 21.4, 18.0. Elemental analysis calcd (%) for C_21_H_17_N_3_: C 81.00, H 5.50, N, 13.49; found: C 81.09, H 5.56, N 13.54.

3-(4-Methoxyphenyl)-6-methylbenzo[4,5]imidazo[1,2-*a*]pyrrolo[2,1-*c*]pyrazine (**3d**). Pink powder (0.144 g, 40% yield; 0.324 g, 90% yield). ^1^H NMR (400 MHz, CDCl_3_) δ 7.90–7.87 (m, 2H, Ph), 7.49 (d, *J* = 8.7 Hz, 2H, Ph), 7.39 (t, *J* = 7.8 Hz, 1H, Ph), 7.35 (d, *J* = 3.9 Hz, 1H, pyrrole), 7.28–7.24 (m, 2H, CH, Ph), 7.06 (d, *J* = 8.7 Hz, 2H, Ph), 6.69 (d, *J* = 3.9 Hz, 1H, pyrrole), 3.89 (s, 3H, CH_3_), 2.81 (s, 3H, CH_3_). ^13^C NMR (100 MHz, CDCl_3_): δ 159.4, 144.5, 143.3, 130.9, 130.4, 129.8 (2C), 123.6, 121.6, 120.6, 120.5, 119.2, 114.4 (2C), 112.4, 112.2, 107.3, 105.9, 55.3, 17.8. Elemental analysis calcd (%) for C_21_H_17_N_3_O: C 77.04, H 5.23, N 12.84; found: C 77.08, H 5.27, N 12.87.

3-(3-Methoxyphenyl)-6-methylbenzo[4,5]imidazo[1,2-*a*]pyrrolo[2,1-*c*]pyrazine (**3e**). White powder (0.137 g, 38% yield; 0.32 g, 89% yield). ^1^H NMR (400 MHz, CDCl_3_) δ 7.90 (m, 2H, Ph), 7.45–7.40 (m, 2H, Ph), 7.38–7.33 (m, 2H, pyrrole, Ph), 7.28–7.25 (m, 1H, CH), 7.16–7.14 (m, 1H, Ph), 7.10 (s, 1H, Ph), 6.99–6.96 (m, 1H, Ph), 6.76 (d, *J* = 3.9 Hz, 1H, pyrrole), 3.88 (s, 3H, CH_3_), 2.81 (s, 3H, CH_3_). ^13^C NMR (100 MHz, CDCl_3_): δ 160.1, 144.8, 143.4, 132.7, 131.2, 130.6, 130.1, 123.9, 121.9, 121.4, 120.99, 120.90, 119.6, 114.5, 113.4, 112.9, 112.6, 107.6, 106.3, 55.5, 18.0. Elemental analysis calcd (%) for C_21_H_17_N_3_O: C 77.04, H 5.23, N 12.84; found: C 77.06, H 5.25, N 12.86.

3-(4-Chlorophenyl)-6-methylbenzo[4,5]imidazo[1,2-*a*]pyrrolo[2,1-*c*]pyrazine (**3f**). White powder (0.124 g, 34% yield; 0.299 g, 82% yield). ^1^H NMR (400 MHz, CDCl_3_) δ 7.90 (d, *J* = 8.1 Hz, 1H, Ph), 7.86 (d, *J* = 8.4 Hz, 1H, Ph), 7.49 (s, 4H, Ph), 7.39 (t, *J* = 7.6 Hz, 1H, Ph), 7.35 (d, *J* = 4.0 Hz, 1H, pyrrole), 7.27–7.25 (m, 1H, Ph), 7.20 (s, 1H, CH), 6.72 (d, *J* = 4.0 Hz, 1H, pyrrole), 2.79 (s, 3H, CH_3_). ^13^C NMR (100 MHz, CDCl_3_): δ 144.8, 143.3, 134.1, 131.2, 129.98, 129.94 (2C), 129.5, 129.4 (2C), 124.0, 122.0, 121.7, 121.2, 119.7, 113.2, 112.6, 107.3, 106.4, 18.1. Elemental analysis calcd (%) for C_20_H_14_ClN_3_: C 72.40, H 4.25, N 12.66; found: C 72.47, H 4.29, N 12.67.

3-(4-Bromophenyl)-6-methylbenzo[4,5]imidazo[1,2-*a*]pyrrolo[2,1-*c*]pyrazine (**3g**). White powder (0.153 g, 37% yield; 0.347 g; 84% yield). ^1^H NMR (400 MHz, CDCl_3_) δ 7.90 (d, *J* = 8.1 Hz, 1H, Ph), 7.87 (d, *J* = 8.3 Hz, 1H, Ph) 7.65 (d, *J* = 8.3 Hz, 2H, Ph), 7.43 (d, *J* = 8.4 Hz, 2H, Ph), 7.39–7.34 (m, 2H, pyrrole, Ph), 7.27–7.26 (m, 1H, Ph), 7.20 (s, 1H, CH), 6.73 (d, *J* = 3.9 Hz, 1H, pyrrole), 2.79 (s, 3H, CH_3_). ^13^C NMR (100 MHz, CDCl_3_): δ 144.7, 143.3, 132.3 (2C), 131.2, 130.4, 130.1 (2C), 129.5, 124.0, 122.2, 122.0, 121.7, 121.2, 119.7, 113.1, 112.6, 107.2, 106.5, 18.0. Elemental analysis calcd (%) for C_20_H_14_BrN_3_: C 63.84, H 3.75, N 11.17; found, %: C 63.91, H 3.82, N 11.20.

6-Methyl-3-(naphthalen-2-yl)benzo[4,5]imidazo[1,2-*a*]pyrrolo[2,1-*c*]pyrazine (**3h**). White powder (0.107 g, 28% yield; 0.302 g, 79% yield). ^1^H NMR (400 MHz, CDCl_3_) δ 8.02–7.97 (m, 2H, Ph), 7.92–7.88 (m, 4H, Ph), 7.68–7.66 (m, 1H, Ph), 7.57–7.54 (m, 2H, Ph), 7.43–7.39 (m, 2H, pyrrole, Ph), 7.37 (s, 1H, CH), 7.29–7.25 (m, 1H, Ph), 6.85 (d, *J* = 4.0 Hz, 1H, pyrrole), 2.80 (s, 3H, CH_3_). ^13^C NMR (100 MHz, CDCl_3_): δ 144.9, 143.5, 133.6, 132.9, 131.3, 130.8, 128.95, 128.91, 128.2, 127.9, 127.5, 126.8, 126.68, 126.61, 124.0, 122.0, 121.6, 121.0, 119.8, 113.4, 112.7, 107.7, 106.6, 18.1. Elemental analysis calcd (%) for C_24_H_17_N_3_: C 82.97, H 4.93, N 12.10; found: C 83.02, H 4.98, N 12.17.

6-Methyl-13,14-dihydrobenzo[*g*]benzo[4’,5’]imidazo[2’,1’:3,4]pyrazino[1,2-*a*]indole (**3i**). White powder (0.121 g, 34% yield; 0.270 g, 76% yield). ^1^H NMR (400 MHz, CDCl_3_) δ 7.90–7.88 (m, 2H, Ph), 7.70–7.65 (m, 2H, Ph), 7.42–7.33 (m, 3H, Ph), 7.28–7.24 (m, 1H, pyrrole), 7.22–7.19 (m, 2H, CH, Ph), 2.97–2.93 (m, 2H, indole), 2.90–2.88 (m, 5H, indole, CH_3_). ^13^C NMR (100 MHz, CDCl_3_): δ 144.8, 143.3, 137.2, 131.2, 129.0, 128.7, 126.8, 126.7, 126.2, 125.9, 123.9, 121.7, 121.3, 120.7, 120.2, 119.5, 112.6, 108.2, 104.6, 30.7, 22.6, 18.1. Elemental analysis calcd (%) for C_22_H_17_N_3_: C 81.71, H 5.30, N 12.99; found: C 81.75, H 5.37; N 13.34.

6-Methyl-3-(thiophen-2-yl)benzo[4,5]imidazo[1,2-*a*]pyrrolo[2,1-*c*]pyrazine (**3j**). White powder (0.137 g, 41% yield; 0.294 g, 88% yield). ^1^H NMR (400 MHz, CDCl_3_) δ7.78 (d, *J* = 8.1 Hz, 1H, Ph), 7.75 (d, *J* = 8.1 Hz, 1H, Ph), 7.32–7.31 (m, 1H, thienyl), 7.30–7.28 (m, 2H, thienyl), 7.22 (d, *J* = 4.0 Hz, 1H, pyrrole), 7.16–7.12 (m, 2H, CH, Ph), 7.09–7.07 (m, 1H, Ph), 6.70 (d, *J* = 4.0 Hz, 1H, pyrrole), 2.69 (s, 3H, CH_3_). ^13^C NMR (100 MHz, CDCl_3_): δ 144.8, 143.1, 132.6, 131.1, 127.8, 126.3, 126.1, 124.0, 123.5, 122.0, 121.7, 121.2, 119.7, 114.1, 112.6, 107.6, 106.3, 18.1. Elemental analysis calcd (%) for C_18_H_13_N_3_S: C 71.26, H 4.32, N 13.85; found: C 71.29, H 4.37, N 13.90.

3-Butyl-5a-methyl-2-propyl-5a,6-dihydro-5*H*,12*H*-benzo[4,5]imidazo[1,2-*a*]pyrrolo[1,2-*d*]pyrazine (**4a**). Orange oil (0.01 g, 4% yield). ^1^H NMR (400 MHz, CDCl_3_) δ 6.79 (m, 1H, Ph), 6.75–6.60 (m, 2H, Ph), 6.39 (d, *J* = 7.6 Hz, 1H, Ph), 5.81 (s, 1H, pyrrole), 4.47 (d, *J* = 14.8 Hz, 1H, CH_2_), 4.44 (d, *J* = 14.8 Hz, 1H, CH_2_), 3.92 (d, *J* = 12.0 Hz, 1H, CH_2_), 3.85 (s, 1H, NH), 3.70 (d, *J* = 12.0 Hz, 1H, CH_2_), 2.50–2.45 (m, 2H, CH_2_), 2.34 (t, *J* = 15.2 Hz, 2H, CH_2_), 1.56–1.50 (m, 2H, CH_2_), 1.43–1.39 (m, 2H, CH_2_), 1.37–1.31 (m, 2H, CH_2_), 1.26 (s, 3H, CH_3_), 0.95–0.90 (m, 6H, CH_3_).

5a-Methyl-3-phenyl-5a,6-dihydro-5*H*,12*H*-benzo[4,5]imidazo[1,2-*a*]pyrrolo[1,2-*d*]pyrazine (**4b**). Red oil (0.102 g, 31% yield). ^1^H NMR (400 MHz, CDCl_3_) δ 7.42–7.35 (m, 4H, Ph), 7.31–7.27 (m, 1H, Ph), 6.80 (t, *J* = 7.5 Hz, 1H, Ph), 6.64 (t, *J* = 7.5 Hz, 1H, Ph), 6.59–6.57 (m, 1H, Ph), 6.46 (d, *J* = 7.5 Hz, 1H, Ph), 6.27 (d, *J* = 3.5 Hz, 1H, pyrrole), 6.12 (d, *J* = 3.5 Hz, 1H, pyrrole), 4.65 (d, *J* = 14.6 Hz, 1H, CH_2_), 4.52 (d, *J* = 14.6 Hz, 1H, CH_2_), 4.16 (d, *J* = 12.1 Hz, 1H, CH_2_), 3.88 (d, *J* = 12.1 Hz, 1H, CH_2_), 3.68 (s, 1H, NH), 1.22 (s, 3H, CH_3_). ^13^C NMR (100 MHz, CDCl_3_): δ 142.5, 138.8, 134.6, 132.5, 128.7 (2C), 128.2 (2C), 126.8, 126.1, 121.6, 119.3, 110.4, 108.2, 107.4, 105.2, 79.9, 52.1, 43.2, 24.7. Elemental analysis calcd (%) for C_20_H_19_N_3_: C 79.70, H 6.35, N 13.94; found: C 79.78, H 6.41, N 13.99.

5a-Methyl-3-(*p*-tolyl)-5a,6-dihydro-5*H*,12*H*-benzo[4,5]imidazo[1,2-*a*]pyrrolo[1,2-*d*]pyrazine (**4c**). Red oil (0.104 g, 30% yield). ^1^H NMR (400 MHz, CDCl_3_) δ 7.31 (d, *J* = 8.0 Hz, 2H, Ph), 7.26 (d, *J* = 8.0 Hz, 2H, Ph), 6.84 (t, *J* = 7.5 Hz,1H, Ph), 6.68 (t, *J* = 7.5 Hz, 1H, Ph), 6.61 (d, *J* = 7.4 Hz, 1H, Ph), 6.49 (d, *J* = 7.5 Hz, 1H, Ph), 6.28 (d, *J* = 3.5 Hz, 1H, pyrrole), 6.15 (d, *J* = 3.5 Hz, 1H, pyrrole), 4.68 (d, *J* = 14.6 Hz, 1H, CH_2_), 4.55 (d, *J* = 14.6 Hz, 1H, CH_2_), 4.17 (d, *J* = 12.1 Hz, 1H, CH_2_), 3.90 (d, *J* = 12.1 Hz, 1H, CH_2_), 3.70 (s, 1H, NH), 2.43 (s, 3H, CH_3_), 1.24 (s, 3H, CH_3_). ^13^C NMR (100 MHz, CDCl_3_): δ 142.4, 138.8, 136.5, 134.6, 129.6, 129.3 (2C), 128.2 (2C), 125.7, 121.6, 119.2, 110.3, 107.8, 107.3, 105.0, 79.9, 52.0, 43.2, 24.7, 21.2. Anal. Calcd for C_21_H_21_N_3_, %: C, 79.97; H, 6.71; N, 13.32. Found, %: C, 80.02; H, 6.74; N, 13.35.

3-(4-Methoxyphenyl)-5a-methyl-5a,6-dihydro-5*H*,12*H*-benzo[4,5]imidazo[1,2-*a*]pyrrolo[1,2-*d*]pyrazine (**4d**). Red oil (0.127 g, 35% yield). ^1^H NMR (400 MHz, CDCl_3_) δ 7.29 (d, *J* = 8.7 Hz, 2H, Ph), 6.95 (d, *J* = 8.6 Hz, 2H, Ph), 6.82–6.78 (m, 1H, Ph), 6.65–6.62 (m, 1H, Ph), 6.59 (d, *J* = 8.8 Hz, 1H, Ph), 6.45 (d, *J* = 7.4 Hz, 1H, Ph), 6.18 (d, *J* = 3.4 Hz, 1H, pyrrole), 6.10 (d, *J* = 3.4 Hz, 1H, pyrrole), 4.64 (d, *J* = 14.7 Hz, 1H, CH_2_), 4.51 (d, *J* = 14.7 Hz, 1H, CH_2_), 4.11 (d, *J* = 12.1 Hz, 1H, CH_2_), 3.84 (s, 3H, CH_3_), 3.80 (d, *J* = 12.1 Hz, 1H, CH_2_), 3.67 (s, 1H, NH), 1.22 (s, 3H, CH_3_). ^13^C NMR (100 MHz, CDCl_3_): δ 158.7, 142.6, 138.8, 134.4, 129.6 (2C), 125.4, 125.1, 121.7, 119.2, 114.1 (2C), 110.4, 107.4, 104.9, 79.9, 77.3, 55.4, 52.0, 43.2, 24.8. Elemental analysis calcd (%) for C_21_H_21_N_3_O: C 76.11, H 6.39, N 12.68; found: C 76.18, H 6.47, N 12.74.

3-(3-Methoxyphenyl)-5a-methyl-5a,6-dihydro-5*H*,12*H*-benzo[4,5]imidazo[1,2-*a*]pyrrolo[1,2-*d*]pyrazine (**4e**). Red oil (0.120 g, 33% yield). ^1^H NMR (400 MHz, CDCl_3_) δ 7.33–7.29 (m, 1H, Ph), 7.26–7.23 (m, 1H, Ph), 6.95–6.93 (m, 1H, Ph), 6.86–6.83 (m, 1H, Ph), 6.82–6.78 (m, 1H, Ph), 6.64 (t, *J* = 7.5 Hz, 1H, Ph), 6.59–5.57 (m, 1H, Ph), 6.46 (d, *J* = 7.5 Hz, 1H, Ph), 6.28 (d, *J* = 3.5 Hz, 1H, pyrrole), 6.12 (d, *J* = 3.5 Hz, 1H, pyrrole), 4.65 (d, *J* = 14.6 Hz, 1H, CH_2_), 4.51 (d, *J* = 14.6 Hz, 1H, CH_2_), 4.15 (d, *J* = 12.1 Hz, 1H, CH_2_), 3.91 (d, *J* = 12.1 Hz, 1H, CH_2_), 3.84 (s, 3H, CH_3_), 3.69 (s, 1H, NH), 1.22 (s, 3H, CH_3_). ^13^C NMR (100 MHz, CDCl_3_): δ 159.8, 142.3, 138.9, 134.5, 133.8, 129.6, 126.2, 121.4, 120.6, 119.3, 113.8, 112.2, 110.1, 108.4, 107.3, 105.2, 79.9, 55.3, 52.1, 43.2, 24.7. Elemental analysis calcd (%) for C_21_H_21_N_3_O: C 76.11, H 6.39, N 12.68. found: C 76.20, H 6.45, N 12.78.

3-(4-Chlorophenyl)-5a-methyl-5a,6-dihydro-5*H*,12*H*-benzo[4,5]imidazo[1,2-*a*]pyrrolo[1,2-*d*]pyrazine (**4f)**. Red oil (0.118 g, 32% yield). ^1^H NMR (400 MHz, CDCl_3_) δ 7.38 (d, *J* = 8.6 Hz, 2H, Ph), 7.30 (d, *J* = 8.6 Hz, 2H, Ph), 6.82 (t, *J* = 7.6 Hz, 1H, Ph), 6.66 (t, *J* = 7.5 Hz, 1H, Ph), 6.61 (d, *J* = 7.4 Hz, 1H, Ph), 6.47 (d, *J* = 7.5 Hz, 1H, Ph), 6.27 (d, *J* = 3.5 Hz, 1H, pyrrole), 6.13 (d, *J* = 3.5 Hz, 1H, pyrrole), 4.65 (d, *J* = 14.7 Hz, 1H, CH_2_), 4.52 (d, *J* = 14.7 Hz, 1H, CH_2_), 4.14 (d, *J* = 12.1 Hz, 1H, CH_2_), 3.82 (d, *J* = 12.1 Hz, 1H, CH_2_), 3.71 (s, 1H, NH), 1.23 (s, 3H, CH_3_). ^13^C NMR (100 MHz, CDCl_3_): δ 142.5, 138.8, 133.4, 132.6, 130.9, 129.3 (2C), 128.9 (2C), 126.5, 121.7, 119.4, 110.5, 108.6, 107.4, 105.4, 79.8, 52.1, 43.1, 24.7. Elemental analysis calcd (%) for C_20_H_18_ClN_3_: C 71.53, H 5.40, N 12.51; found: C 71.59, H 5.48, N 12.55.

3-(4-Bromophenyl)-5a-methyl-5a,6-dihydro-5*H*,12*H*-benzo[4,5]imidazo[1,2-*a*]pyrrolo[1,2-*d*]pyrazine (**4g**). Red oil (0.138 g, 33% yield). ^1^H NMR (400 MHz, CDCl_3_) δ 7.53 (d, *J* = 8.4 Hz, 2H, Ph), 7.23 (d, *J* = 8.4 Hz, 2H, Ph), 6.83 (t, *J* = 7.4 Hz, 1H, Ph), 6.65 (t, *J* = 7.0 Hz, 1H, Ph), 6.60 (d, *J* = 7.5 Hz, 1H, Ph), 6.46 (d, *J* = 7.3 Hz, 1H, Ph), 6.27 (d, *J* = 3.5 Hz, 1H, pyrrole), 6.13 (d, *J* = 3.5 Hz, 1H, pyrrole), 4.64 (d, *J* = 14.7 Hz, 1H, CH_2_), 4.51 (d, *J* = 14.7 Hz, 1H, CH_2_), 4.13 (d, *J* = 12.1 Hz, 1H, CH_2_), 3.82 (d, *J* = 12.1 Hz, 1H, CH_2_), 3.69 (s, 1H, NH), 1.22 (s, 3H, CH_3_). ^13^C NMR (100 MHz, CDCl_3_): δ 142.4, 138.7, 133.4, 131.8 (2C), 131.4, 129.6 (2C), 126.6, 121.7, 120.7, 119.4, 110.5, 108.7, 107.4, 105.4, 79.8, 52.1, 43.1, 24.7. Elemental analysis calcd (%) for C_20_H_18_BrN_3_: C 63.17, H 4.77, N 11.05; found: C 63.23, H 4.85, N 11.13.

5a-Methyl-3-(naphthalen-2-yl)-5a,6-dihydro-5*H*,12*H*-benzo[4,5]imidazo[1,2-*a*]pyrrolo[1,2-*d*]pyrazine (**4h**). Red oil (0.073 g, 19% yield). ^1^H NMR (400 MHz, CDCl_3_) δ 7.89–7.83 (m, 3H, Ph), 7.79 (s, 1H, Ph), 7.55–7.46 (m, 3H, Ph), 6.83 (t, *J* = 7.5 Hz, 1H, Ph), 6.69 (t, *J* = 7.5 Hz, 1H, Ph), 6.60 (d, *J* = 7.4 Hz, 1H, Ph), 6.49 (d, *J* = 7.5 Hz, 1H, Ph), 6.41 (d, *J* = 3.5 Hz, 1H, pyrrole), 6.19 (d, *J* = 3.5 Hz, 1H, pyrrole), 4.69 (d, *J* = 14.6 Hz, 1H, CH_2_), 4.56 (d, *J* = 14.6 Hz, 1H, CH_2_), 4.26 (d, *J* = 12.2 Hz, 1H, CH_2_), 3.97 (d, *J* = 12.2 Hz, 1H, CH_2_), 3.67 (s, 1H, NH), 1.22 (s, 3H, CH_3_). ^13^C NMR (100 MHz, CDCl_3_): δ 142.5, 138.8, 134.7, 133.6, 132.2, 129.9, 128.3, 127.9, 127.8, 126.7, 126.5, 126.4, 126.2, 125.9, 121.7, 119.3, 110.4, 108.8, 107.4, 105.5, 79.9, 52.4, 43.3, 24.8. Elemental analysis calcd (%) for C_24_H_21_N_3_: C 82.02, H 6.02, N 11.96; found: C 82.14, H 6.16, N 12.12.

14a-Methyl-5,8,14a,15-tetrahydro-6*H*,14*H*-benzo[*g*]benzo[4’,5’]imidazo[1’,2’:4,5]pyrazino[1,2-*a*]indole (**4i**). Red oil (0.1 g, 28% yield). ^1^H NMR (400 MHz, CDCl_3_) δ 7.29–7.24 (m, 2H, Ph), 7.21–7.18 (m, 1H, Ph), 7.06 (t, *J* = 7.4 Hz, 1H, Ph), 6.82 (t, *J* = 7.5 Hz, 1H, Ph), 6.68–6.62 (m, 1H, Ph), 6.44 (d, *J* = 7.7 Hz, 1H, Ph), 5.97 (s, 1H, pyrrole), 4.62 (d, *J* = 14.7 Hz, 1H, CH_2_), 4.55 (d, *J* = 14.7 Hz, 1H, CH_2_), 4.35 (d, *J* = 11.9 Hz, 1H, CH_2_), 4.20 (d, *J* = 11.9 Hz, 1H, CH_2_), 3.75 (s, 1H, NH), 2.95–2.82 (m, 2H, indole), 2.72–2.58 (m, 2H, indole), 1.29 (s, 1H, CH_3_). ^13^C NMR (100 MHz, CDCl_3_): δ 142.5, 138.7, 136.1, 129.5, 129.0, 128.6, 126.6, 126.1, 124.6, 122.7, 121.9, 120.3, 119.1, 110.8, 107.0, 103.8, 79.5, 77.3, 55.3, 42.8, 31.1, 24.4, 22.5. Elemental analysis calcd (%) for C_22_H_21_N_3_: C 80.70, H 6.46, N 12.83; found: C 80.78, H 6.51, N 12.90.

5a-Methyl-3-(thiophen-2-yl)-5a,6-dihydro-5*H*,12*H*-benzo[4,5]imidazo[1,2-*a*]pyrrolo[1,2-*d*]pyrazine (**4j**). Red oil (0.125 g, 37% yield). ^1^H NMR (400 MHz, CDCl_3_) δ 7.29 (d, *J* = 3.6 Hz, 1H, thienyl), 7.10–7.08 (m, 1H, thienyl), 6.98 (d, *J* = 3.5 Hz, 1H, thienyl), 6.82 (t, *J* = 7.4 Hz, 1H, Ph), 6.67 (m, 1H, Ph), 6.62 (d, *J* = 7.2 Hz, 1H, Ph), 6.47 (d, *J* = 7.5 Hz, 1H, Ph), 6.35 (d, *J* = 3.5 Hz, 1H, pyrrole), 6.11 (d, *J* = 3.5 Hz, 1H, pyrrole), 4.63 (d, *J* = 15.1 Hz, 1H, CH_2_), 4.55 (d, *J* = 15.1 Hz, 1H, CH_2_), 4.11 (d, *J* = 12.2 Hz, 1H, CH_2_), 4.02 (d, *J* = 12.2 Hz, 1H, CH_2_), 3.74 (s, 1H, NH), 1.33 (s, 1H, CH_3_). ^13^C NMR (100 MHz, CDCl_3_): δ 142.1, 138.7, 134.3, 127.5, 126.6, 126.2, 124.9, 124.7, 121.6, 119.2, 110.5, 109.6, 107.1, 105.0, 79.6, 51.5, 42.5, 24.3. Elemental analysis calcd (%) for C_18_H_17_N_3_S: C 70.33, H 5.57, N, 13.67; found: C 70.45, H 5.67, N 13.76.

2-(5-Butyl-4-propyl-1H-pyrrol-2-yl)-1*H*-benzo[*d*]imidazole (**5a**). Yellow powder (0.157 g, 28% yield). ^1^H NMR (400 MHz, DMSO-*d_6_*) δ 12.25 (s, 1H, NH), 11.18 (s, 1H, NH), 7.43 (s, 2H, Ph), 7.08–7.05 (m, 2H, Ph), 6.60 (s, 1H, pyrrole), 2.51–2.47 (m, 2H, CH_2_), 2.31 (t, *J* = 7.4 Hz, 2H, CH_2_), 1.54–1.45 (m, 4H, CH_2_), 1.26–1.19 (m, 2H, CH_2_), 0.89 (t, *J* = 7.4 Hz, 3H, CH_3_), 0.84 (t, *J* = 7.3 Hz, 3H, CH_3_). ^13^C NMR (100 MHz, DMSO-*d_6_*): δ 147.2, 132.7, 121.24, 121.4, 120.3, 119.9, 106.6, 32.2, 27.5, 24.8, 24.1, 21.9, 13.9, 13.8. Elemental analysis calcd (%) for C_18_H_23_N_3_: C 76.83, H 8.24, N 14.93; found: C 76.96, H 8.31, N 15.03.

2-(4,5,6,7-Tetrahydro-1*H*-indol-2-yl)-1*H*-benzo[*d*]imidazole (**5l**). Yellow powder (0.175 g, 37% yield). ^1^H NMR (400 MHz, DMSO-*d_6_*) δ 12.27 (s, 1H, NH), 11.20 (s, 1H, NH), 7.85–7.38 (m, 2H, Ph), 7.10–7.08 (m, 2H, Ph), 6.55 (s, 1H, pyrrole), 2.57–2.55 (m, 2H, indole), 2.47–2.42 (m, 2H, indole), 1.74–1.67 (m, 2H, indole). ^13^C NMR (100 MHz, DMSO-*d_6_*): δ 177.6, 147.3, 130.8, 121.4, 121.3, 121.1, 120.5, 117.6, 110.4, 108.1, 30.75, 23.5, 22.9, 22.63, 22.62. Elemental analysis calcd (%) for C_15_H_15_N_3_: C 75.92, H 6.37, N 17.71; found: C 76.03, H 6.48, N 17.82.

2-(5-(*p*-Tolyl)-1*H*-pyrrol-2-yl)-1*H*-benzo[*d*]imidazole (**5c**). Beige powder (0.41 g, 75% yield). ^1^H NMR (400 MHz, DMSO-*d_6_*) δ 12.48 (s, 1H, NH), 11.96 (s, 1H, NH), 7.76 (d, *J* = 8.0 Hz, 2H, Ph), 7.59–7.49 (m, 2H, Ph), 7.20–7.16 (m, 4H, Ph), 6.95 (d, *J* = 3.2 Hz, 1H, pyrrole), 6.65 (d, *J* = 3.2 Hz, 1H, pyrrole), 2.30 (s, 3H, CH_3_). ^13^C NMR (100 MHz, DMSO-*d_6_*): δ 146.4, 143.9, 135.6, 134.8, 134.5, 129.2 (2C), 124.4 (2C), 123.8, 121.7, 121.3, 117.8, 111.1, 110.6, 107.1, 20.7. Elemental analysis calcd (%) for C_18_H_15_N_3_: C 79.10, H 5.53, N 15.37; found: C 79.15, H 5.58, N 15.41.

2-(5-(4-Methoxyphenyl)-1*H*-pyrrol-2-yl)-1*H*-benzo[*d*]imidazole (**5d**). Pink powder (0.121 g, 21% yield). ^1^H NMR (400 MHz, DMSO-*d_6_*) δ 12.48 (s, 1H, NH), 11.90 (s, 1H, NH), 7.79 (d, *J* = 8.7 Hz, 2H, Ph), 7.57–7.47 (m, 2H, Ph), 7.16–7.14 (m, 2H, Ph), 6.97 (d, *J* = 8.7 Hz, 2H, Ph), 6.92 (d, *J* = 3.6 Hz, 1H, pyrrole), 6.57 (d, *J* = 3.6 Hz, 1H, pyrrole), 3.77 (s, 3H, CH_3_). ^13^C NMR (100 MHz, DMSO-*d_6_*): δ 146.7, 144.2, 135.9, 135.2, 134.8, 129.6 (2C), 124.7 (2C), 124.2, 122.1, 121.6, 118.1, 111.4, 107.5, 21.1. Elemental analysis calcd (%) for C_18_H_15_N_3_O: C 74.72, H 5.23, N 14.52: found: C 74.77, H 5.29, N 14.58.

2-(5-(3-Methoxyphenyl)-1*H*-pyrrol-2-yl)-1*H*-benzo[*d*]imidazole (**5e**). Red powder (0.231 g, 40% yield). ^1^H NMR (400 MHz, DMSO-*d_6_*) δ 12.53 (s, 1H, NH), 12.09 (s, 1H, NH), 7.59 (m, 1H, Ph), 7.51 (m, 1H, Ph), 7.48–7.47 (m, 1H, Ph), 6.40 –7.38 (m, 1H, Ph), 7.28 (t, *J* = 7.9 Hz, 1H, Ph), 7.17–7.16 (m, 2H, Ph), 6.94 (d, *J* = 3.0 Hz, 1H, pyrrole), 6.79–6.77 (m, 1H, Ph), 6.72 (d, *J* = 3.0 Hz, 1H, pyrrole), 3.83 (s, 3H, CH_3_). ^13^C NMR (100 MHz, DMSO-*d_6_*): δ 146.0, 133.4, 131.4, 131.2, 126.3 (2C), 124.6, 121.6 (2C), 119.1, 114.2, 111.2, 108.3. Elemental analysis calcd (%) for C_18_H_15_N_3_O: C 74.72, H 5.23, N 14.52; found: C 74.87, H 5.36, N 14.65.

2-(5-(4-Chlorophenyl)-1*H*-pyrrol-2-yl)-1*H*-benzo[*d*]imidazole (**5f**). Brown powder (0.487 g, 83% yield). ^1^H NMR (400 MHz, DMSO-*d_6_*) δ 12.57 (s, 1H, NH), 12.10 (s, 1H, NH), 7.87 (d, *J* = 8.5 Hz, 2H, Ph), 7.59 (s, 1H, Ph), 7.47 (s, 1H, Ph), 7.44 (d, *J* = 8.5 Hz, 2H, Ph), 7.17–7.16 (m, 2H, Ph), 6.96 (d, *J* = 3.7 Hz, 1H, pyrrole), 6.73 (d, *J* = 3.7 Hz, 1H, pyrrole). ^13^C NMR (100 MHz, DMSO-*d_6_*): δ 146.2, 133.5, 130.9, 130.8, 128.8 (2C), 126.1 (2C), 124.6, 122.1, 121.6, 118.0, 111.3, 110.8, 108.4. Elemental analysis calcd (%) for C_17_H_12_ClN_3_: C 69.51, H 4.12, N 14.30; found: C 69.64, H 4.21, N 14.42.

2-(5-(4-Bromophenyl)-1*H*-pyrrol-2-yl)-1*H*-benzo[*d*]imidazole (**5g**). Light yellow powder (0.277 g, 41% yield). ^1^H NMR (400 MHz, DMSO-*d_6_*) δ 12.66 (s, 1H, NH), 12.13 (s, 1H, NH), 7.84 (d, *J* = 8.4 Hz, 2H, Ph), 7.57–7.53 (m, 4H, Ph), 7.18–7.16 (m, 2H, Ph), 6.96 (d, *J* = 3.2 Hz, 1H, pyrrole), 6.74 (d, *J* = 3.2 Hz, 1H, pyrrole). ^13^C NMR (100 MHz, DMSO-*d_6_*): δ 146.0, 133.4, 131.4, 131.2, 126.3 (2C), 124.6, 121.6 (2C), 119.1, 114.2, 111.2, 108.3. Elemental analysis calcd (%) for C_17_H_12_BrN_3_: C 60.37, H 3.58, N 12.42; found: C 60.42, H 4.05, N 12.47.

2-(5-(Naphthalen-2-yl)-1*H*-pyrrol-2-yl)-1*H*-benzo[*d*]imidazole (**5h**). Beige powder (0.185 g, 30% yield). ^1^H NMR (400 MHz, DMSO-*d_6_*) δ 12.54 (s, 1H, NH), 12.18 (s, 1H, NH), 8.48 (s, 1H, Ph), 8.02–7.99 (m, 1H, Ph), 7.92–7.87 (m, 3H, Ph), 7.63–7.61 (m, 1H, Ph), 7.53–7.44 (m, 3H, Ph), 7.19–7.17 (m, 2H, Ph), 7.00 (d, *J* = 3.2 Hz, 1H, pyrrole), 6.87 (d, *J* = 3.2 Hz, 1H, pyrrole). ^13^C NMR (100 MHz, DMSO-*d_6_*): δ 146.2, 143.9, 134.59, 134.52, 133.4, 131.7, 129.4, 128.0, 127.7, 127.5, 126.4, 125.5, 124.6, 123.5, 121.9, 121.8, 121.3, 117.8, 111.1, 110.7, 108.4. Elemental analysis calcd (%) for C_21_H_15_N_3_: C 81.53, H, 4.89, N 13.58; found: C 81.58, H 4.95, N 13.64.

2-(1*H*-Benzo[*d*]imidazol-2-yl)-4,5-dihydro-1*H*-benzo[*g*]indole (**5i**). Beige powder (0.313 g, 55% yield). ^1^H NMR (400 MHz, DMSO-*d_6_*) δ 12.43 (s, 1H, NH), 12.05 (s, 1H, NH), 7.85 (d, *J* = 8.5 Hz, 1H, Ph), 7.49 (s, 2H, Ph), 7.16–7.11 (m, 4H, Ph), 7.04–7.00 (m, 1H, Ph), 6.75 (s, 1H, pyrrole), 2.84 (t, *J* = 7.3 Hz, 2H, indole), 2.67 (t, *J* = 7.3 Hz, 2H, indole). ^13^C NMR (100 MHz, DMSO-*d_6_*): δ 146.5, 134.7, 130.4, 128.9, 127.9, 126.6, 125.4, 123.2, 121.4, 120.8, 120.7, 108.6, 29.4, 21.3. Elemental analysis calcd (%) for C_19_H_15_N_3_: C 79.98, H 5.30, N 14.73; found: C 80.02, H 5.36, N 14.77.

5-Methylbenzo[4,5]imidazo[1,2-*a*]pyrrolo[2,1-*c*]pyrazine (**6k**). Beige powder (0.199 g, 82% yield). ^1^H NMR (400 MHz, CDCl_3_) δ 7.89 (d, *J* = 8.4 Hz, 1H, Ph), 7.61 (d, *J* = 7.9 Hz, 1H, Ph), 7.43–7.39 (m, 2H, Ph, pyrrole), 7.34–7.30 (m, 1H, pyrrole), 7.28–7.27 (m, 2H, CH, pyrrole), 6.79–6.78 (m, 1H, pyrrole), 2.53 (s, 3H, CH_3_). ^13^C NMR (100 MHz, CDCl_3_): δ 143.8, 142.0, 129.6, 123.8, 121.6, 121.4, 120.1, 119.2, 115.7, 112.8, 109.1, 106.8, 105.4, 15.6. Elemental analysis calcd (%) for C_14_H_11_N_3_: C 76.00, H 5.01, N 18.99: found: C 76.12, H 5.17, N 19.13.

3-Butyl-5-methyl-2-propylbenzo[4,5]imidazo[1,2-*a*]pyrrolo[2,1-*c*]pyrazine (**6a**). Yellow powder (0.214 g, 61% yield). ^1^H NMR (400 MHz, CDCl_3_) δ 7.71 (d, *J* = 8.0 Hz, 1H, Ph), 7.43 (d, *J* = 8.0 Hz, 1H, Ph), 7.24 (t, *J* = 7.6 Hz, 1H, Ph), 7.15–7.13 (m, 2H, Ph, CH), 6.92 (s, 1H, pyrrole), 2.89–2.85 (m, 2H, CH_2_), 2.59 (s, 3H, CH_3_), 2.44 (t, *J* = 7.5 Hz, 2H, CH_2_), 1.62–1.56 (m, 2H, CH_2_), 1.49 –1.45 (m, 2H, CH_2_), 1.36–1.30 (m, 2H, CH_2_), 0.92–0.84 (m, 6H, CH_3_). ^13^C NMR (100 MHz, CDCl_3_): δ 144.4, 142.7, 130.7, 129.7, 127.3, 123.8, 121.3, 121.1 (2C), 119.0, 108.9, 107.2, 106.2, 34.5, 28.4, 25.9, 23.8, 22.6, 18.6, 14.2, 13.9. Elemental analysis calcd (%) for C_21_H_25_N_3_: C 78.96, H 7.89, N 13.15; found: C 79.05, H 7.97, N 13.21.

6-Methyl-1,2,3,4-tetrahydrobenzo[4’,5’]imidazo[2’,1’:3,4]pyrazino[1,2-*a*]indole (**6l**). Beige powder (0.154 g, 51% yield). ^1^H NMR (400 MHz, CDCl_3_) δ 7.80 (d, *J* = 8.1 Hz, 1H, Ph), 7.44 (d, *J* = 8.0 Hz, 1H, Ph), 7.44 (t, *J* = 7.6 Hz, 1H, Ph), 7.33 (t, *J* = 7.6 Hz, 1H, Ph), 7.21 (t, *J* = 7.6 Hz, 1H, Ph), 7.06 (s, 1H, CH), 6.82 (s, 1H, pyrrole), 3.11 (t, *J* = 6.0 Hz, 2H, indole), 2.69 (t, *J* = 5.5 Hz, 2H, indole), 2.54 (s, 3H, CH_3_), 1.88–1.84 (m, 2H, indole), 1.77–1.74 (m, 2H, indole). ^13^C NMR (100 MHz, CDCl_3_): δ 144.3, 142.5, 129.7, 128.2, 124.1, 123.6, 121.7, 121.1, 119.0, 108.9, 106.0, 105.2, 77.36, 25.9, 24.0, 23.8, 22.7, 18.6. Elemental analysis calcd (%) for C_18_H_17_N_3_: C 78.52, H 6.22, N 15.26; found: C 78.65, H 6.35, N 15.36.

5-Methyl-3-phenylbenzo[4,5]imidazo[1,2-*a*]pyrrolo[2,1-*c*]pyrazine (**6b**). Brown powder (0.264 g, 81% yield). ^1^H NMR (400 MHz, CDCl_3_) δ 88 (d, *J* = 8.1 Hz, 1H, Ph), 7.59 (d, *J* = 7.9 Hz, 1H, Ph), 7.48–7.46 (m, 2H, Ph), 7.45–7.38 (m, 5H, Ph, pyrrole), 7.30 (t, *J* = 7.5 Hz, 1H, Ph), 7.10 (s, 1H, CH), 6.65 (d, *J* = 3.9 Hz, 1H, pyrrole), 2.03 (s, 3H, CH_3_). ^13^C NMR (100 MHz, CDCl_3_): δ 144.1, 142.7, 134.3, 133.1 131.1 (2C), 129.6, 128.6, 127.7 (2C), 124.0, 122.8, 122.0, 121.6, 119.2, 115.9, 109.1, 106.8, 106.7, 19.6. Elemental analysis calcd (%) for C_20_H_15_N_3_: C 80.78, H 5.08, N 14.13; found: C 80.83, H 5.14, N 14.19.

5-Methyl-3-(*p*-tolyl)benzo[4,5]imidazo[1,2-*a*]pyrrolo[2,1-*c*]pyrazine (**6c**). Yellow powder (0.291 g, 85% yield). ^1^H NMR (400 MHz, CDCl_3_) δ 7.91 (d, *J* = 8.1 Hz, 1H, Ph), 7.51 (d, *J* = 7.9 Hz, 1H, Ph), 7.44 (d, *J* = 3.9 Hz, 1H, pyrrole), 7.40–7.36 (m, 3H, Ph), 7.28–7.24 (m, 3H, Ph), 6.95 (s, 1H, CH), 6.63 (d, *J* = 3.9 Hz, 1H, pyrrole), 2.48 (s, 3H, CH_3_), 2.00 (s, 3H, CH_3_). ^13^C NMR (100 MHz, CDCl_3_): δ 144.1, 142.6, 138.4, 133.0, 131.1, 130.8 (2C), 129.5, 128.2 (2C), 123.7, 122.5, 121.8, 121.3, 119.0, 115.5, 109.0, 106.5, 106.3, 21.3, 19.4. Elemental analysis calcd (%) for C_21_H_17_N_3_: C 81.00, H 5.50, N 13.49; found: C 81.13, H 5.58, N 13.60.

3-(4-Methoxyphenyl)-5-methylbenzo[4,5]imidazo[1,2-*a*]pyrrolo[2,1-*c*]pyrazine (**6d**). Brown powder (0.323 g, 90% yield). ^1^H NMR (400 MHz, CDCl_3_) δ 7.86 (d, *J* = 8.0 Hz, 1H, Ph), 7.55 (d, *J* = 8.1 Hz, 1H, Ph), 7.40 (d, *J* = 3.9 Hz, 1H, pyrrole), 7.38 (d, *J* = 8.5 Hz, 2H, Ph), 7.29–7.25 (m, 2H, Ph), 7.03 (s, 1H, CH), 6.94 (d, *J* = 8.7 Hz, 2H, Ph), 6.59 (d, *J* = 3.9 Hz, 1H, pyrrole), 3.87 (s, 3H, CH_3_), 2.00 (s, 3H, CH_3_). ^13^C NMR (100 MHz, CDCl_3_): δ 159.9, 144.1, 142.7, 132.8, 132.2 (2C), 129.6, 126.3, 123.8, 122.5, 121.9, 121.4, 119.1, 115.6, 113.0 (2C), 109.0, 106.6, 106.4, 55.4, 19.4. Elemental analysis calcd (%) for C_21_H_17_N_3_O: C 77.04, H 5.23, N 12.84; found: C 77.10, H 5.28, N 12.91.

3-(3-Methoxyphenyl)-5-methylbenzo[4,5]imidazo[1,2-a]pyrrolo[2,1-c]pyrazine (**6e**). Brown powder (0.317 g, 88% yield). ^1^H NMR (400 MHz, CDCl_3_) δ 7.76 (d, *J* = 8.1 Hz, 1H, Ph), 7.44 (d, *J* = 8.1 Hz, 1H, Ph), 7.30 (d, *J* = 3.9 Hz, 1H, pyrrole), 7.27–7.25 (m, 1H, Ph), 7.22–7.14 (m, 2H, Ph), 6.93–6.86 (m, 4H, CH, Ph), 6.52 (d, *J* = 3.9 Hz, 1H, pyrrole), 3.73 (s, 3H, CH_3_), 2.95 (s, 3H, CH_3_). ^13^C NMR (100 MHz, CDCl_3_): δ 158.9, 144.2, 142.7, 135.5, 132.8, 129.7, 128.6, 123.9, 123.7, 122.9, 121.9, 121.6, 119.3, 116.8, 115.7, 114.2, 109.1, 106.8, 106.5, 55.4, 19.3. Elemental analysis calcd (%) for C_21_H_17_N_3_O: C 77.04, H 5.23, N 12.84; found: C 77.10; H 5.28, N 12.91.

3-(4-Chlorophenyl)-5-methylbenzo[4,5]imidazo[1,2-*a*]pyrrolo[2,1-*c*]pyrazine (**6f**). Brown powder (0.303 g, 83% yield). ^1^H NMR (400 MHz, CDCl_3_) δ 7.89 (d, *J* = 8.1 Hz, 1H, Ph), 7.61 (d, *J* = 8.0 Hz, 1H, Ph), 7.43–7.39 (m, 6H, Ph, pyrrole), 7.32 (t, *J* = 7.6 Hz, 1H, Ph), 7.13 (s, 1H, CH), 6.63 (d, *J* = 3.9 Hz, 1H, pyrrole), 2.05 (s, 3H, CH_3_). ^13^C NMR (100 MHz, CDCl_3_): δ 144.2, 142.5, 134.7, 132.7, 132.2 (2C), 131.5, 129.6 (2C), 127.9, 124.1, 123.1, 121.7, 121.6, 119.4, 116.1, 109.1, 107.0, 106.7, 19.8. Elemental analysis calcd (%) for C_20_H_14_ClN_3_: C 72.40, H 4.25, N 12.66; found: C 72.49, H 4.31, N 12.74.

3-(4-Bromophenyl)-5-methylbenzo[4,5]imidazo[1,2-*a*]pyrrolo[2,1-*c*]pyrazine (**6g**). Yellow powder (0.397 g, 96% yield). ^1^H NMR (400 MHz, CDCl_3_) δ 7.89 (d, *J* = 8.1 Hz, 1H, Ph), 7.59–7.56 (m, 3H, Ph), 7.42–7.39 (m, 1H, pyrrole), 7.36 (d, *J* = 8.0 Hz, 2H, Ph), 7.33–7.27 (m, 2H, Ph), 7.10 (s, 1H, CH), 6.63 (d, *J* = 3.9 Hz, 1H, pyrrole), 2.04 (s, 3H, CH_3_). ^13^C NMR (100 MHz, CDCl_3_): δ 144.1, 142.4, 133.1, 132.4 (2C), 131.4, 130.8 (2C), 129.5, 124.0, 123.1, 122.8, 121.6, 121.5, 119.2, 116.0, 109.1, 106.9, 106.6, 19.7. Elemental analysis calcd (%) for C_20_H_14_BrN_3_: C 63.84, H 3.75, N 11.17; found, %: C 63.95, H 3.81, N 11.28.

5-Methyl-3-(naphthalen-2-yl)benzo[4,5]imidazo[1,2-*a*]pyrrolo[2,1-*c*]pyrazine (**6h**). Light brown powder (0.302 g, 79% yield). ^1^H NMR (400 MHz, CDCl_3_) δ 7.97 (s, 1H, Ph), 7.93–7.87 (m, 4H, Ph), 7.61–7.56 (m, 4H, Ph), 7.48 (d, *J* = 3.9 Hz, 1H, pyrrole), 7.41 (t, *J* = 7.3 Hz, 1H, Ph), 7.32 (t, *J* = 7.4 Hz, 1H, Ph), 7.13 (s, 1H, CH), 6.73 (d, *J* = 3.9 Hz, 1H, pyrrole), 2.06 (s, 3H, CH_3_). Elemental analysis calcd (%) for C_24_H_17_N_3_: C 82.97, H 4.93, N 12.10; found: C 83.11, H 5.07, N 12.19.

7-Methyl-13,14-dihydrobenzo[*g*]benzo[4’,5’]imidazo[2’,1’:3,4]pyrazino[1,2-*a*]indole (**6i**). Brown powder (0.299 g, 84% yield). ^1^H NMR (400 MHz, CDCl_3_) δ 7.89 (d, *J* = 7.9 Hz, 1H, Ph), 7.65 (d, *J* = 7.9 Hz, 1H, Ph), 7.42 (t, *J* = 7.4 Hz, 1H, Ph), 7.35–7.26 (m, 5H, Ph, pyrrole), 7.19–7.16 (m, 2H, Ph, CH), 2.95 (t, *J* = 6.7 Hz, 2H, indole), 2.79 (t, *J* = 6.7 Hz, 2H, indole), 2.58 (s, 3H, CH_3_). ^13^C NMR (100 MHz, CDCl_3_): δ 144.4, 142.7, 136.8, 130.6, 129.9, 129.6, 129.3, 128.0, 125.9, 125.8, 125.1, 124.5, 124.1, 121.9, 121.7, 119.4, 109.3, 108.1, 105.9, 30.8, 23.4, 19.8. Elemental analysis calcd (%) for C_22_H_17_N_3_: C 81.71, H 5.30, N 12.99; found: C 81.78, H 5.39, N 13.07.

### 3.4. X-ray Crystallography

Crystal Data for 3e: Formula: C_21_H_17_N_3_O·CHCl_3_ (M = 327.38 g/mol): monoclinic, space group P21/c (No. 14), a = 10.3012(10) Å, b = 22.0073(19) Å, c = 7.2627(7) Å, β = 90.469(3)°, V = 1566.72(9) Å^3^, Z = 4, T = 296(2) K, µ(CuKα) = 1.076 mm^−1^, Dcalc = 1.261 g/cm^3^, 43,389 reflections measured (2.30° ≤ θ ≤ 30.06°), 4583 unique (Rint = 0.1327; Rsigma = 0.0865), which were used in all calculations. The final R_1_ was 0.0699 (I > 2σ(I)) and wR_2_ was 0.1920 (all data). CCDC 2102328 contains the supplementary crystallographic data for this paper. The data can be obtained free of charge from The Cambridge Crystallographic Data Centre via http://www.ccdc.cam.ac.uk (accessed on 10 August 2021).

Crystal Data for 6b: Formula: C_20_H_15_N_3_·CHCl_3_ (M = 297.35 g/mol): monoclinic, space group P21/c (No. 14), a = 10.4033(3) Å, b = 20.1257(7) Å, c = 7.8873(3) Å, β = 108.427(2)°, V = 1646.4(3) Å^3^, Z = 4, T = 296(2) K, µ(CuKα) = 0.083 mm^−1^, Dcalc = 1.321 g/cm^3^, 42,575 reflections measured (3.36° ≤ θ ≤ 30.05°), 4804 unique (Rint = 0.0288; Rsigma = 0.0186), which were used in all calculations. The final R_1_ was 0.0555 (I > 2σ(I)) and wR_2_ was 0.0774 (all data). CCDC 2102327 contains the supplementary crystallographic data for this paper. The data can be obtained free of charge from The Cambridge Crystallographic Data Centre via http://www.ccdc.cam.ac.uk (accessed on 10 August 2021).

## 4. Conclusions

In conclusion, three different series of valuable and highly promising annulated heterocyclic assemblies have been synthesized from available building blocks, pyrrole/indolecarbaldehyde, *o*-phenylenediamine and propargyl chloride, by simply controlling the water content of the reaction. Moreover, 5a-methyl-5a,6-dihydro-5*H*,12*H*-benzo[4,5]imidazo[1,2-*a*]pyrrolo[1,2-*d*] **4** and 5-methylbenzo[4,5]imidazo[1,2-*a*]pyrrolo[2,1-*c*]pyrazine **6** with the methyl group in *α*-position to pyrrole were obtained for the first time. These compounds represent important scaffolds for pharmaceutical chemistry, given the wide biological activity of nitrogen-containing heterocycles and manifold possibilities of their further functionalization.

## Data Availability

The data presented in this study are available in the Appendix A.

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
