# Peer review of "Solvent Moisture-Controlled Self-Assembly of Fused Benzoimidazopyrrolopyrazines with Different Ring’s Interposition"

_molecules, 2022, doi:10.3390/molecules27082460_

Round 1
Reviewer 1 Report
The article “Humidity-Controlled Self-Assembly of Fused Benzoimidazopyrrolopyrazines with Different Ring’s Interposition” reports the synthesis of 6-methylbenzo[4,5]imidazo[1,2-a]pyrrolo[2,1-c]pyrazine and 5a-methyl-5a,6-dihydro-5H,12H-benzo[4,5]imidazo[1,2-a]pyrrolo[1,2-d]pyrazine by one step. Reaction conditions are optimized to obtain the 6-methylbenzo[4,5]imidazo[1,2-a]pyrrolo[2,1-c]pyrazine as major product. It is also reported the structural characterization of all synthesized compounds.
The authors present the results in table 2, pay attention to entry 1. Correct “c” by “a”.
The first reaction mechanism for the synthesis of compound 3, needs to be supported by literature reference. Scheme 5 should be reorganized to make the proposed mechanism more perceptible. How Baldwin’s rule apply in the nine-membered ring formation?
The authors present the synthesis of compound H and the structural characterization, but the 1H NMR spectrum (SI data) appear to be a mixture of the starting material ((5-phenyl-1-(propa-1,2-dien-1-yl)-1H-pyrrol-2-yl)methanol) and compound H. The authors also report the HMBC correlations, but no HMBC spectrum was present.
In terms of compound 6 synthesis.
Please, introduce the reaction time at the step 2.
The authors could reorganize the order of compound 6 at the table 3. First appear compound k then a and l…I don´t understand these organization, normally the organization is compound a, b, c,…
At the General procedure for synthesis of 6, the authors use “freshly distilled propargyl chloride”. The authors could provide a briefly description of the propargyl chloride distillation at Materials and Methods?
Author Response
Response to Reviewer 1 Comments
Point 1: The authors present the results in table 2, pay attention to entry 1. Correct “c” by “a”
Response 1: Corrected as recommended
Point 2: The first reaction mechanism for the synthesis of compound 3, needs to be supported by literature reference. Scheme 5 should be reorganized to make the proposed mechanism more perceptible.
Response 2: The relevant reference supporting the first reaction mechanism was given. Scheme 5 was reorganised as advised.
Point 3: How Baldwin’s rule apply in the nine-membered ring formatio.
Response 3: In accordance with the literature data, Baldwin's rule can be applied to the formation of macrocycles including nine-membered ones. The following references were included to the manuscript: Org. Biomol. Chem., 2019, 17, 8806–8810; Angew. Chem. Int. Ed. 2014, 53, 3409 –3413; WIREs Computational Molecular Science 2016, 6, 487-514.
Point 4: The authors present the synthesis of compound H and the structural characterization, but the 1H NMR spectrum (SI data) appear to be a mixture of the starting material ((5-phenyl-1-(propa-1,2-dien-1-yl)-1H-pyrrol-2-yl)methanol) and compound H. The authors also report the HMBC correlations, but no HMBC spectrum was present.
Response 4: Indeed, in the case of compound H, a mixture of the starting compound and the product was formed, which was indicated in the supporting information. HMBC spectrum was added.
Point 5: In terms of compound 6 synthesis. Please, introduce the reaction time at the step 2.
Response 5: The reaction time was introduced at the step 2.
Point 6: The authors could reorganize the order of compound 6 at the table 3. First appear compound k then a and l…I don´t understand these organization, normally the organization is compound a, b, c,..
Response 6: Compounds k and l appear only in the synthesis of 6. This is a cause of such numbering. In our opinion, the numbering of compounds in accordance with substituents will be confusing for the first part of the paper.
Point 7: At the General procedure for synthesis of 6, the authors use “freshly distilled propargyl chloride”. The authors could provide a briefly description of the propargyl chloride distillation at Materials and Methods?
Response 7: Done as recommended
Reviewer 2 Report
Suggestions for corrections are given in the attached file.

Author Response
Response to Reviewer 2 Comments
Point 1: Extensive editing of English language and style required.
Response 1: Mistakes were corrected.
Point 2: This manuscript (ID: molecules - 1654462) describes an interesting study that leads to two valuable ring systems. The findings should be of interest to organic, heterocyclic and medicinal chemists. I have a few suggestions for the authors that will make their paper better. First, "humidity" refers to the concentration of water in the air, not in solvent. Thus, I would refer to this as "water content of the medium". The title should be changed to "Solvent-Controlled Self-Assembly of Fused Benzoimidazopyrrolopyrazines with Different Ring Positions". This title also breaks up the long compound name in the correct position and changes a little of the wording at the end.
Response 2: The title was changed as advised: "Solvent Moisture-Controlled Self-Assembly of Fused Benzoimidazopyrrolopyrazines with Different Ring Positions”.
Point 3: Figure 2 would be clearer if there were two structures, one with NOESY and the other with HMBC correlations.
Response 3: Figure 2 was edited as recommended.
Point 4: In Scheme 6, where does the H2O lost in going from H to I come from in structure H?
Response 4: Mistakes were corrected.
Point 5: The reference cited should be indicated by [20] and listed in the reference section.” “...literature method. (The reference for drying ethanol and benzene should be indicated by [21] and listed in the reference section.)
Response 5: The references were cited in the reference section.
Point 6: DMSO-d6 is used for deuterated DMSO. ”δ in the 13C NMR data does not have to be followed by an equal (=) sign.
Response 6: Coorrected.
Point 7: “..pyrrolodihydropyrazine[1,2-d]benzimidazoles 4 and pyrrolopyrazine[1,2-d]benzimidazoles 6 with the methyl group α to the pyrrole were obtained for the first time. Why are the compounds named differently here? Keep the nomenclature consistent throughout the manuscript.
Response 7: The compounds named were corrected.
Point 8: Some of the spectra in the supplemental section have some impurities, especially compounds 4 and 5 (there is no 13C for 4a). Most of the TMS peaks (supposed to be at δ 0.00) line up with 0 (one spectrum says TMS is at δ 1.14). I also note that the solvent peak used as the standard is δ 77.16 rather than δ 77.10 as it says in the General Experimental Information..
Response 8: The spectra of compounds 4 and 5 have some impurities due to the difficulties with purification of reaction mixture using column chromatography. Unfortunately, the repeated columnt purification gave no better results. Other purification methods did not result in the desired purity of the product. Indeed, there is no 13C spectrum for 4a, because the yield of the product turned was extremely low and the amount of the obtained compound was insufficient to record the 13C spectrum. Solvent peak δ 77.10 was corrected to δ 77.16 in the General Experimental Information.
Round 2
Reviewer 2 Report
The authors have made most of the corrections I suggested. I noticed a few small things that the authors or editors may want to correct.
line 25: leukocyte transport (this error may have been mine)
line 67: ...contains some water...
line 170: Supplemental Information or SI
In the experimental, DMSO-d6 should have the 6 subscripted.
Otherwise, the paper is ready to be published.
Author Response
Point 1: leukocyte transport (this error may have been mine)
Response 1: Mistake was corrected.
Point 2: …contains some water…
Response 2: Mistake was corrected.
Point 3: Supplemental Information or SI
Response 3: Mistake was corrected.
Point 4: In the experimental, DMSO-d6 should have the 6 subscripted.
Response 4: Mistake was corrected.